# On the Effectiveness of Quasi Character-Level Models for Machine Translation

## Abstract

Neural Machine Translation (NMT) models often use subword-level vocabularies to deal with rare or unknown words. Although some studies have shown the effectiveness of purely character-based models, these approaches have resulted in highly expensive models in computational terms. In this work, we explore the advantages of quasi character-level Transformers for low-resource NMT, as well as their ability to mitigate the catastrophic forgetting problem. We first present an empirical study on the effectiveness of these models as a function of the size of the training set. As a result, we found that for data-poor environments, quasi character-level Transformers present a competitive advantage over their large subword-level versions. Similarly, we study the generalization of this phenomenon in different languages, domains, and neural architectures. Finally, we conclude this work by studying the ability of these models to mitigate the effects of catastrophic forgetting in machine translation. Our work suggests that quasi character-level Transformers have a competitive advantage in data-poor environments and, although they do not mitigate the catastrophic forgetting problem, they greatly help to achieve greater consistency between domains.

## 1 Introduction

Neural machine translation (NMT) has become the dominant paradigm in the field of machine translation due to the impressive results achieved with encoder-decoder architectures (Sutskever et al. (2014); Cho et al. (2014a); Wu et al. (2016); Vaswani et al. (2017)).

Despite these advances, the translation of rare or unknown words has turned out to be a more difficult problem to tackle than was initially thought. For which reason, authors have proposed different approaches to deal with this problem, which can be grouped into three categories: i) Models based on character-level representations. ii) Hybrid NMT Models. iii) Models based on word segmentation.

Character-based models are the direct approach as they can naturally deal with rare or unseen words. However, these models have historically resulted in unsatisfactory results (Vilar et al. (2007); Neubig et al. (2013)) or highly expensive models in computational terms (Luong & Manning (2016b)). To balance the benefits of word- and character-based models, Hybrid NMT models were invented (Luong & Manning (2016a)). The idea behind these models is to translate mainly at the word level and only query character components for rare words when necessary. However, these models tend to be a bit cumbersome as they generally need two models to do the back-off. Finally, word segmentation approaches such as BPE (Sennrich et al. (2016)), Unigram (Kudo (2018)), or SentencePiece (Kudo & Richardson (2018)), emerged to address many of the problems mentioned above, by allowing the encoding of rare words using a fixed-vocabulary of subwords.

Despite the success of word segmentation models, it is still not clear how to determine the vocabulary size. The most common strategy is to choose an arbitrarily large value (i.e. 32k), under the premise that large vocabularies tend to work well (Gowda & May (2020)). Despite this heuristic, we think that there must be a better way of choosing this value depending on the dataset properties.

Motivated by these ideas, we decided to study whether NMT character-based models had any advantages compared to models with large subword vocabularies (i.e 32k tokens). We theorize that since words follow a Zipfian distribution, models with large vocabularies would require large datasets to learn the long-tailed distribution of subwords. In contrast, character-based models would be able

to learn it more easily in data-poor environments as their subword distribution is less pronounced than the distribution for large vocabularies. One of the main criticisms of character-level models is that they involve longer sequences and therefore, more long-term dependencies. To alleviate some of these problems, we decided to study quasi character-level models, which are essentially small subword-level vocabularies (i.e 100-400 tokens).

Furthermore, we decided to study if character-based models can mitigate the effects of catastrophic forgetting based on the idea that, if this phenomenon depends on the vocabulary's domain, when a vocabulary is shared between domains, these effects should be mitigated.

The contributions of this paper are fourfold:

- Quasi character-level Transformers seem to consistently outperform large subword-level Transformers on low-resource environments. (Section 5.1.1).

- When LSTMs and CNNs are used, this phenomenon is still present but it is more sensitive on the chosen hyperparameters. (Section 5.1.3).

- Quasi character-level Transformers appear to be more susceptible to catastrophic forgetting than their large subword-level counterparts. (Section 5.2).

- However, quasi character-level Transformers achieved better cross-domain consistencies. (Section 5.2).

## 2 RELATED WORK

Character-based models have been well-studied in the Natural Language Processing (NLP) field to deal with the open-vocabulary problem. One of the first character-based models was proposed by Vilar et al. (2007), who treated the source and target sentences as a string of letters. Similarly, Neubig et al. (2013) viewed translation as a single transduction between character strings in the source. However, despite the relevance of their contributions, their results were not satisfactory as their models generally performed worse than their word-based counterparts.

Consequently, authors began to propose strategies to get the best of word- and character-level models. These approaches were mostly based on Hybrid NMT models or subword-level models with vocabularies built through word segmentation techniques. Luong & Manning (2016a) achieved open vocabulary NMT through hybrid models, which translate at the word level and consult the character components only for rare words; and Sennrich et al. (2016) developed a model to segment words based on the Byte-Pair encoding compression algorithm proposed by Gage (1994). Later, more word segmentation models were proposed based on this idea, such as Unigram by Kudo (2018) or SentencePiece by Kudo & Richardson (2018).

Chung et al. (2016) demonstrated that a NMT model with a character-based decoder can outperform NMT models with subword-level decoders. Similarly, Luong & Manning (2016b) and Costa-jussà & Fonollosa (2016) showed that competitive purely character-based NMT models were possible. Despite their relative success, these models were extremely slow to train and at runtime.

Recurrent neural architectures have been the dominant neural architecture for NLP tasks (Cho et al. (2014b); Wu et al. (2016); Sundermeyer et al. (2014)). However, in recent years we have seen convolutional neural models outperform recurrent models on some NLP tasks. Gehring et al. (2017) proposed a competitive sequence to sequence architecture based entirely on convolutional neural networks, and Conneau et al. (2016) achieved state-of-the-art machine translation with a purely character-level NMT model via deep convolutional stacks.

Recently, the attention mechanism (Bahdanau et al. (2015); Luong et al. (2015)) along with the Transformer architecture by Vaswani et al. (2017), has led to significant progress in the field of NLP, due to the parallelizable nature of the Transformer architectures and its ability to deal with long-term dependencies.

This paper ends with a brief discussion on the ability of quasi character-based Transformers to mitigate the catastrophic forgetting problem in machine translation. As far as we know, this is the first work to address this issue from the perspective of (quasi) character-level models, since most of the existing alternatives are based on regularization (i.e. LwF by Li & Hoiem (2016), EWC

by Kirkpatrick et al. (2016)), dynamic architectures (Rusu et al. (2016); Draelos et al. (2016)) or Complementary Learning Systems (CLS) and Memory Replay (MR) (Kemker & Kanan (2017); Lopez-Paz & Ranzato (2017)).

## 3 Neural Machine Translation

### 3.1 Neural architectures for Machine Translation

The goal of any translation system is to transform an input sequence in a given language into an output sequence in a target language.

Nowadays, this is usually done using neural models based on the encoder-decoder architecture; also know as seq2seq models in the machine translation community (Sutskever et al. (2014)). The encoder part transforms the input sequence into an internal representation, and then the decoder transforms this internal representation into the output sequence.

In this work, we focus our study on the Transformer architecture, although recurrent and fully convolutional architectures are also explored for completeness.

Recurrent architectures (RNNs) were the first to be successfully applied in an encoder-decoder setup for machine translation. Even though there are many types of RNNs, most of them chain sequentially a series of unit cells to process temporal sequences. We decided to use LSTMs (Hochreiter & Schmidhuber (1997)) because their units cells are explicitly designed to deal with long-term dependencies.

Convolution-based architectures (CNN) do not contain any recurrent elements. They can do this, because the idea behind this architecture is that the convolutional filters can slide through the sequence of tokens, from beginning to end (Gehring et al. (2017)). Initially, the convolutional layers were intended to operate on 2-dimensional images so, in order to apply them to text data, we have to imagine the sequence of tokens as if it were a *1xm* image, where the height of the filter is equal to *1*, and the width equal to *k*. Being *k* the number of consecutive tokens that the filter can observe at a given time.

Finally, we have Transformer architecture by Vaswani et al. (2017), a state-of-the-art architecture based entirely on the concept of *attention* (Bahdanau et al. (2015); Luong et al. (2015)) to draw global dependencies between the input and output. This architecture does not use any recurrence or convolutional layer to process temporal sequences. Instead, it makes use of linear and normalization layers, residual connections, attention mechanisms, and some tricks to encode temporal information.

### 3.2 The open vocabulary problem

In the written language it is common to find alternative spellings (i.e. color-colour) and typos (i.e acknowledge-acknowlege) that slightly modify the spelling of a word but do not prevent us, the humans, from knowing its meaning. However, if a model is using a word-level representation, it will stop knowing a *known word* at the very first moment that it is slightly modified and this modification is not in its vocabulary. Furthermore, it must be taken into account that most languages make use of agglutination and compounding mechanisms to form new words, which in itself makes it very inefficient to build a vocabulary with all possible words of a language.

Because of this, researchers have proposed multiple approaches to deal with the open vocabulary problem. These approaches can be mostly grouped into three categories: i) Character-based models ii) Hybrid models iii) Subword-based models.

Character-based models contain the minimum possible vocabulary with which to form all possible words. Consequently, these models can translate rare or even unseen words character-by-character, but at the same time, these models tend to be much slower and harder to train than word-based models. This is because they have to deal with longer sequences and therefore, longer long-term dependencies.

Hybrid models tend to translate primarily at word-level but fall back to character-level translation when a rare or unseen word appears.

Finally, subword-based models are fixed close-vocabularies that use subwords instead of full words. These models allow us to represent infrequent or even unknown words as a sequence of subwords, but without the performance and memory issues that typically imply character-based models.

In this work, we always work at the subword level but distinguishing two types of subword vocabularies. The first one is the quasi character-level vocabulary with 100-400 tokens, and the second one is a large subword-level vocabulary with around 32,000 tokens.

To build our subword-level vocabularies we chose a word segmentation model known as Byte-Pair Encoding (BPE) by Sennrich et al. (2016). We did also consider Unigram (Kudo (2018)) and SentencePiece (Kudo & Richardson (2018)) but eventually, we decided to use BPE due to its popularity. The idea behind BPE is to start with a set of characters and iteratively extend it with the most frequent n-grams pairs in the corpus.

## 4 EXPERIMENTAL SETUP

### 4.1 DATASETS

The data used in this work comes mostly from the WMT tasks. In table 1 we find all the datasets used in this work.

Table 1: Datasets partitions

| Dataset | Train | Val | Test |
|---|---|---|---|
| **Europarl v7 (ES-EN)** | 1.9M / 100k / 50k | 5000 | 5000 |
| **Europarl v7 (DE-EN)** | 1,8M / 100k / 50k | 5000 | 5000 |
| **Europarl v7 (CS-EN)** | 635k / 100k / 50k | 5000 | 5000 |
| **CommonCrawl (ES-EN)** | 1.8M / 100k | 5000 | 5000 |
| **SciELO (ES-EN)** | 575k / 120k | 4961 | 4961 |
| **SciELO (PT-EN)** | 116k | 3738 | 3738 |
| **NewsCommentary (DE-EN)** | 357k / 35k | 5000 | 5000 |
| **IWLST'16 (DE-EN)** | 196k | 993 | 1305 |
| **Multi30K (DE-EN)** | 29k | 1014 | 1000 |

All the values in this Table indicate the number of sentences.

Europarl contains parallel sentences extracted from the European Parliament website; CommonCrawl consists of parallel texts mined from the Common Crawl dataset; SciELO is a parallel corpus made up of scientific publications for the biomedical domain (health and biological); NewCommentary contains political and economic commentaries, crawled from the website Project Syndicate; Multi30k is a small dataset from WMT 2016 multimodal task, also known as *Flickr30k*; and the IWLST'16 dataset comes from the IWSLT 2016 TED talk translation task.

In addition to this, we created two additional versions for each training set, with 100k and 50k sentences (randomly sampled).[1] The validation and test sets were left untouched.

### 4.2 TRAINING DETAILS

First, we normalized and tokenized each dataset with Sacremoses, a Python port of the Moses tokenizer by Koehn et al. (2007). Then, we created two vocabularies for each dataset with different subword granularities. The first one was made up of 32k subwords, while the other was limited to 100-400 subwords (characters + common subwords)[2]. Both were built using FastBPE, a tool to learn and apply BPE codes. The large subword-level vocabulary was achieved using 32k merge operations, while the small subword-level vocabulary only had 64 merge operations.

We used Fairseq v1.0.0a0 (Ott et al. (2019)) as our sequence modeling toolkit. For our experimentation, we use the following base architectures: *LSTMModel*, *FConvModel* and *TransformerModel*.[3]

---

[1]Later we added more versions (120k, 35k, etc) to perform additional experiments

[2]The exact amount of tokens depended on each dataset

[3]Long Short-Term Memory model (LSTMModel), Fully convolutional model (FConvModel), and the Transformer model (TransformerModel)

In addition to this, we tried two versions of the Transformer architecture: the Standard Transformer with 93M parameters and a smaller version with 28M parameters (see Table 2). Interestingly, both versions performed quite similarly in terms of accuracy (±1 BLEU). Also, the smaller version was notably faster than the standard version, so we end up using the smaller version for most of our experimentation.

Regarding the comparison of the models, we tried to make the comparison as fair a possible by only comparing models with a similar number of parameters (see Table 2). Similarly, we shared as many of the training hyperparameters as possible in order to reduce training biases. Nonetheless, this will be discussed in detail in see Section 5.1.3, as this comparison is tricky due to the impact of the chosen hyperparameters on the final result.

Table 2: Neural architectures

| Model | Parameters | Hyperparameters |
|---|---|---|
| **LSTM** | 6.8M (S) — 27.8M (L) | 3 layers / 256 emb / 256 hid / bidir+attn |
| **Fully CNN** | 6.5M (S) — 27.4M (L) | 6x(256, 3) (enc/dec) |
| **Transformer small** | 4.1M (S) — 25.0M (L) | 3 layers / 8 heads / 256 dim / 512 ffnn |
| **Transformer standard** | 44.9M (S) — 92.7 (L) | 6 layers / 8 heads / 512 dim / 2048 ffnn |

**Common hyperparamters:** Loss function: CrossEntropy (without label smoothing). Optimizer: Adam (Kingma & Ba (2015)) or NAG. Batch size: 4096 tokens/batch, Clip-norm: 0.1. Maximum epochs: 100 epochs with early stopping. Beam width: 5.
**Note:** *S/L* stands for *Small/Large* subword vocabulary

### 4.3 EVALUATION METRICS

NMT models are generally evaluated qualitatively and quantitatively. The qualitative part is carried out by humans, while the quantitative evaluation is carried out with automatic metrics such as BLEU (Papineni et al. (2002)), chrF (Popović (2015)), or TER (Snover et al. (2006)).

These automatic metrics compute the quality of a model by comparing its output with a reference translation written by a human. To do so, we computed the BLEU, chrF, and TER scores using Sacrebleu (Post (2018)) with its default arguments.[4]

## 5 RESULTS

### 5.1 ON THE EFFECTIVENESS OF CHARACTER-BASED MODELS

In this section, we propose a series of experiments to demonstrate that under a series of conditions, quasi character-based models can obtain significant advantages over models based on large vocabularies.

Words follow a Zipfian distribution, which implies that in a large enough corpus, the most frequent word will occur twice as often as the second most frequent word, three times as often as the third most frequent word, and so on. Starting from this idea, we theorized that models with large subword vocabularies would require large datasets to learn the long-tailed distribution of subwords (see Figure 1). In contrast, we thought that models with small subword vocabularies would learn their subword distribution more easily in data-poor environments than their counterparts because since the long-tail distribution of small subword-level vocabularies is shorter, the average relative frequency per token is should be higher which would lead to more opportunities to learn each token in a more homogeneous distribution. Similarly, we argue that the advantages of character-based models should be lost when there is enough data from which to learn the low-frequent subwords of the tail due to the issues related to the longer long-tail dependencies of character-based models.

---

[4]Even though we evaluated each model using BLEU, chrF, and TER, all of them behaved extremely similarly, so we end up reporting just the BLEU metric for clarity.

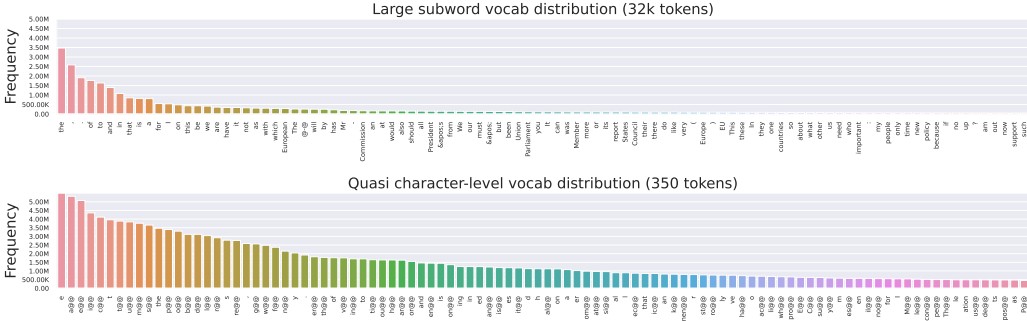

Figure 1: The frequency of the first 100 tokens of a large subword vocabulary (32k tokens) and a small subword vocabulary (350 tokens). The first distribution has a much longer tail than the second distribution, which translates to a lower average relative frequency per token (see first Figure).

### 5.1.1 ADVANTAGES OF TRANSFORMERS FOR DATA-POOR ENVIRONMENTS

To test our hypothesis, we created smaller versions of our datasets by limiting the amount of training data to 100k and 50k sentences. As expected, we see in Figure 2 that as we limit the size of a training set, the performance of quasi character-level Transformer increases compared to their large subword-level versions. The right column corresponds to the full datasets for the Spanish-English, German-English, and Czech-English pairs, while the middle and left columns correspond to the same datasets but limited to 100k and 50k sentences. Our experimentation shows that for each of the original datasets, the Transformers with large subword vocabularies outperform every quasi character-level Transformer by around 5 points in BLEU. However, when we trained them with the smaller versions of those datasets (middle and left columns), every quasi character-level Transformer outperformed each of their large subword-level versions.

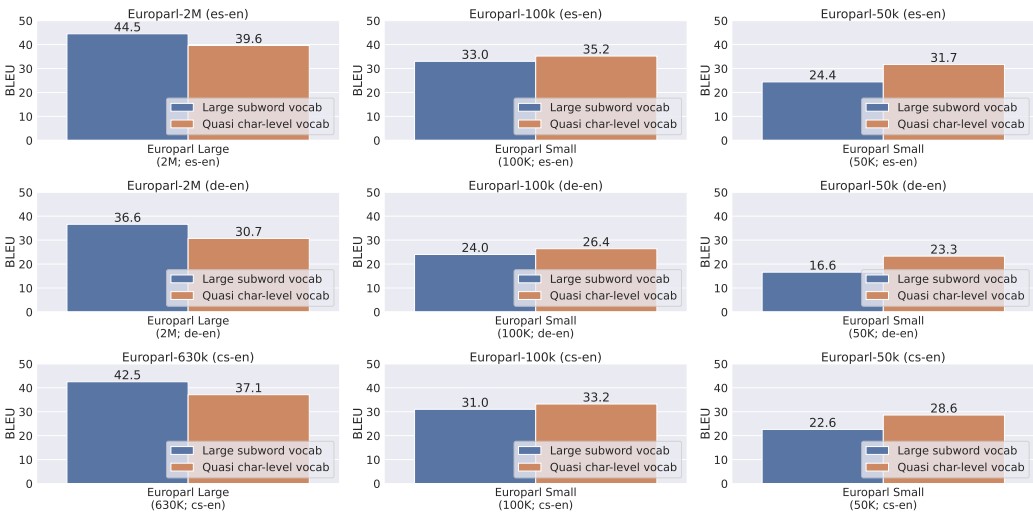

Figure 2: As we limit the training data (left to right), we see that quasi character-level Transformers performs better than their large subword vocabularies versions. Similarly, this phenomenon seems to occur regardless of the language (top to bottom).

### 5.1.2 LANGUAGE AND DOMAIN GENERALIZATION

Initially, we performed the previous experiment on the Spanish-English pair, but in order to account for potential language biases, we decided to repeat the experiment but using different languages.

Our go-to alternative language was German since it is well-known for its propensity for nominal compounding and agglutination. Similarly, we added the Czech-English pair to our experimentation as it keeps more distance with the other two pairs (see Figure 2), as well as other pairs for completeness (i.e. Portuguese-English, Spanish-English). This phenomenon was consistent across all language pairs we tested, which could imply that this is a language-agnostic phenomenon. Nonetheless, we would like to repeat these experiments with more distant pairs (e.g. Chinese-Russian, Japanese-Spanish, etc.) so that we could confidently assert that this is indeed a language-agnostic phenomenon.

Since all the previous datasets came from the Europarl domain, we wanted to explore whether this phenomenon could be caused by domain bias. To do so, we repeated the same experiment but on different domains, such as crawled data, political and economic news, health and biological sciences, transcripted talks, and multimodal transcriptions. These domains came from the following datasets: CommonCrawl, NewsCommentary, SciELO (health and biological datasets), IWLST'16, and Multi30k. See Figure 3.

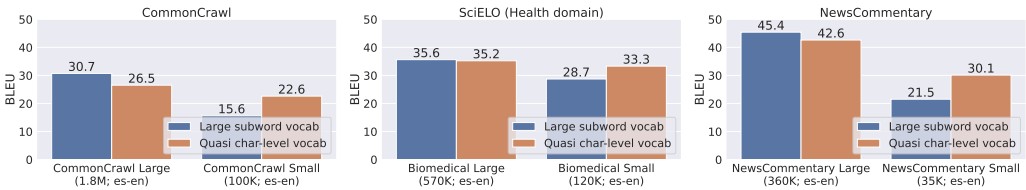

Figure 3: The advantages of quasi character-level Transformers in data-poor environments are consistent when models are trained on different languages and domains. From left to right we have the results from models trained on the CommonCrawl, SciELO, and NewCommentary datasets, which belong to different domains. In each of those figures, we see that when the training data is limited (two columns on the left of each figure), quasi character-level models outperform their large subword-level counterparts, regardless of the domain.

As we see in Figure 3, the phenomenon here studied is present when we change the training domain to crawled data (CommonCrawl), scientific literature (SciELO), and, political and economic commentaries (NewsCommentaries). In addition to these domains, we also experiment with transcripted TED talks (IWLST'16) and image descriptions (Multi30k). As expected, quasi character-level models kept outperforming their large subword-level versions when we limited the training sets. They improved the BLEU score in 6.2pts for the IWLST'16 dataset and 2.3pts for the Multi30k dataset.

### 5.1.3 NEURAL ARCHITECTURES GENERALIZATION

As all the previous studies were done using the Transformer architecture, so we decided to study whether the phenomenon described above generalizes to other neural architectures for machine translation, such as LSTMs or CNNs. Specifically, we focused our research on bidirectional LSTMs with attention mechanisms and fully convolutional architectures like the one described in Gehring et al. (2017). To have a fairer comparison, we only compared models that had a similar number of parameters for a given vocabulary (i.e. 25-30M parameters for vocabularies of 32k subwords).

Since the performance of a model does not depend solely on the number parameters but on its architecture, how it was trained, etc. We decided to standardize the training hyperparameters we used amongst all models as much as possible. In addition to this, we performed multiple runs for each model with slight variations in their hyperparameters and initializations to account for this performance variability.

From our experimentation, we observed that as expected, when large subword-level models were trained with enough data (unlimited training sets), they outperformed their quasi character-level counterparts. This happened for all three architectures (LSTM, CNN, and Transformer). However, when the training sets were limited to 100k or 50k sentences, quasi character-based models performed better than their large subword-level versions in most cases (see Figure 4).

When Transformers were used, the ones with quasi character-level vocabularies consistently outperformed the ones with large subword-level vocabularies. In contrast, when LSTMs or CNNs,

their performance was more dependent on their hyperparameters. In the case of LSTMs, the quasi character-level versions always end up outperforming the large subword vocabularies, but the improvements were not as impressive as the ones seen with the Transformer architecture. Finally, we found CNN-based architectures to be particularly sensitive to the chosen hyperparameters, so the advantages of quasi character-level CNNs are not as clear as to us compared to the other models. Nevertheless, with enough computational time and the right hyperparameters, quasi character-CNNs probably end up outperforming consistently their large subword counterparts.

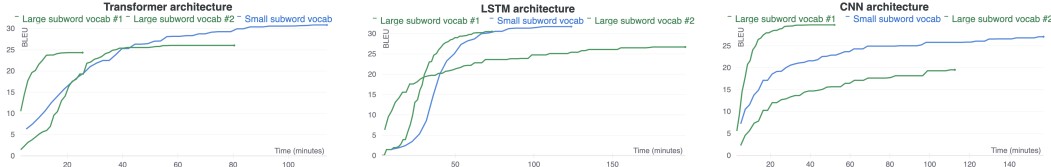

Figure 4: The green lines refer to the best and worst runs of the models with large subword-level vocabularies, while the blue lines refer to the best run of the quasi character-level models. In the left figure, we see that quasi character-level Transformers consistently outperform the ones with large subword vocabularies. Then, in the middle figure we see that for the LSTMs, this phenomenon is not as strong as with Transformers. Finally, in the right figure, we can see how quasi character-CNNs do not present these advantages as clearly as in the other models.

## 5.2 EFFECTS OF QUASI CHARACTER-BASED TRANSFORMERS ON THE CATASTROPHIC FORGETTING PROBLEM

In this section, we study whether quasi character-based Transformers can help to mitigate the catastrophic forgetting problem, whereby neural networks forget previously learned information after learning new information.

To do this, we have designed an experiment in which we train a model in a domain $A$ and evaluate it in domains $A$ and $B$ to establish the baselines. Next, we fine-tune the model trained in domain $A$ with data from the new domain $B$, and then, it is evaluate it in domains $A$ and $B$. In theory, the model trained in domain $A$ should perform well in the domain $A$, and poorly in the unseen domain $B$. Similarly, after fine-tune it on domain $B$, it should perform worse in $A$ and better in domain $B$ than the original model trained only on domain $A$.

In Figure 5a we see that quasi character-level Transformer trained on the health domain (SciELO) obtained a BLEU of 33.3pts on its domain (health) and a BLEU of 14.3pts in the other domain (biological). Then, when we fine-tune this model on the Biological domain (SciELO), the BLEU obtained on this domain increased from 14.3 to 31.7pts, while BLEU for the health domain fell from 33.3 to 21.0pts. In Figure5b we see that something similar happened for the large subword-vocabulary, but the effects of the catastrophic forgetting problem were not as strong as in the other model, because the BLEU score went from 28.7 to 28.0pts.

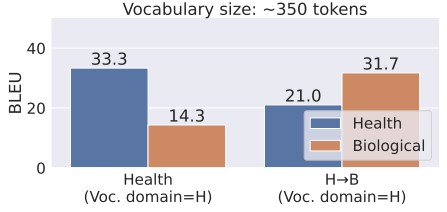
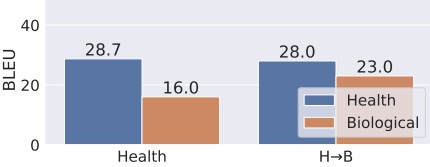

| (a) Quasi character-level vocabulary | (b) Large subword-level vocabulary |

Figure 5: Vocabularies seem to have a strong impact on the catastrophic forgetting effects. While the quasi character-level model lost 12.3pts (see Figure 5a), the large subword-level model only lost 0.7pts (see Figure 5b)

From Figure 5, we can infer that vocabularies seem to have a strong impact on the effects of catastrophic forgetting because character-level vocabularies seem to make models more susceptible to the catastrophic forgetting problem than large subword-level vocabularies.

To further explore this problem, we repeated the previous experiment but taking into account the vocabulary domain. As a result, we discovered that the vocabulary domain has a stronger impact on the model's performance than we thought. As shown in Figure 6, quasi character-level Transformers present a high consistency between domains, while large subword-level Transformers seem particularly sensitive to their vocabulary's domain, to the point of achieving opposite results between domains (see right column of the Figure 6).

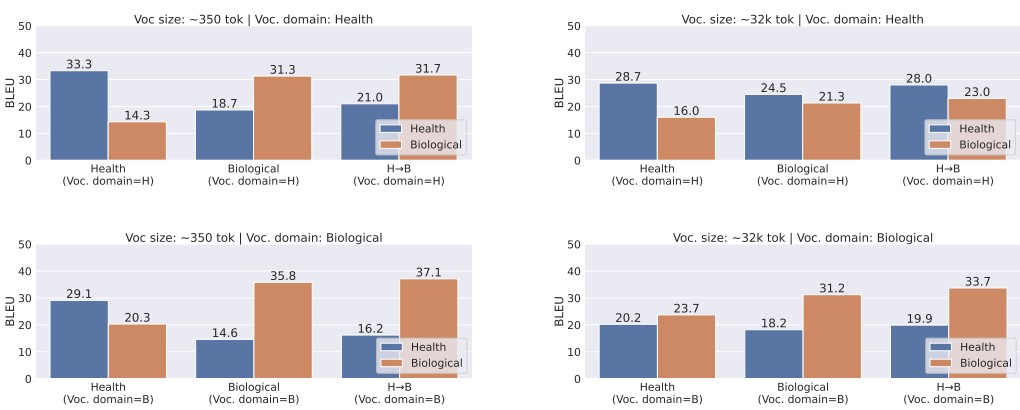

Figure 6: Transformers with quasi character-level vocabularies (left figures) appear to be more consistent between domains than Transformers with large subword-level vocabularies (right figures)

Even though quasi character-level Transformers achieve better consistency across domains, they suffer more severely the catastrophic forgetting problem than Transformers with large subword vocabularies. We believe that by using specially designed regularization techniques to address this issue such as LwF (Li & Hoiem (2016)) or EWC (Kirkpatrick et al. (2016)) these problems could be mitigated, leading to more robust and consistent models.

## 6 CONCLUSION

In this paper, we have empirically studied the effectiveness of quasi character-level models and their ability to tackle the catastrophic forgetting problem.

Our studies reveal that for low-resource scenarios, quasi character-level Transformers are superior in terms of accuracy compared to Transformers with large subword vocabularies. Although this phenomenon seems to generalize with language and domain, it also appears to be dependent on neural architecture. When LSTMs were used, the improvements were not as impressive as with the Transformer architecture, but when we experimented with CNNs, we noticed that the outcome was more dependent on the chosen hyperparameters than with the other models.

Finally, we showed that even though quasi character-level vocabularies do not seem to mitigate the effects of the catastrophic forgetting problem, they achieved a higher consistency between domains, which might lead to substantial improvements if regularization techniques are applied to deal with catastrophic forgetting.

ACKNOWLEDGMENT

Work supported by the Horizon 2020 - European Commission (H2020) under the SELENE project (grant agreement no 871467) and the project Deep learning for adaptive and multimodal interaction in pattern recognition (DeepPattern) (grant agreement PROMETEO/2019/121). We gratefully acknowledge the support of NVIDIA Corporation with the donation of a GPU used for part of this research.

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
