# OpenReview forum: "On the Effectiveness of Quasi Character-Level Models for Machine Translation"
_ICLR.cc/2022/Conference — ICLR 2022 Submitted_

### Official Review · Reviewer_K79J · 2021-11-02

**Correctness:** 3
**Technical Novelty And Significance:** 2
**Empirical Novelty And Significance:** 2
**Recommendation:** 5
**Confidence:** 4

**Main Review:**

The paper is easy to read and follow. However, I think this paper is missing a crucial comparison, of using character-based NMT to compare against the proposed method. Also, the vocabulary size of 350 feels arbitrary, similar to your point of "32k vocab size being arbitrary". I'd suggest having some experiments trying different vocabulary sizes and comparing the quality.

Moreover, the authors criticize the character-based NMT to be slow and inefficient. However, none of the experiments is designed to benchmark the efficiency. I.e. comparison of the average tokens after tokenization with standard subword vs quasi-character vs character-based. I think this experiment will give a good insight into the performance between approaches.

Your claim about "it is still not clear how to determine the vocabulary size." is not entirely true. Prior research ([Optimizing Transformer for Low-Resource Neural Machine Translation](https://aclanthology.org/2020.coling-main.304/) and [Revisiting low-resource neural machine translation: A case study](https://arxiv.org/pdf/1905.11901)) show that a lower vocabulary size is better for low-resource NMT, which is the same conclusion as your paper. However, you did not mention anything about their study.


**Summary Of The Paper:**

The authors explore using a significantly lower vocabulary to improve low-resource NMT. They also experiment with different languages, domains, and architectures.

**Summary Of The Review:**

This paper is only comparing 2 approaches: standard subword tokenization vs the proposed quasi-character. The paper is missing several critical comparisons, for example, character-based NMT, and quasi-character on different vocabulary sizes. There is also no report about the performance, despite the author criticizing the performance issue about character-based NMT. Lastly, the paper is also missing related works that have the same claim of low-vocabulary size is better for low resource NMT.

---

### Official Review · Reviewer_5PDf · 2021-11-02

**Correctness:** 2
**Technical Novelty And Significance:** 2
**Empirical Novelty And Significance:** 3
**Recommendation:** 3
**Confidence:** 5

**Main Review:**

Strengths:
- This paper has a well-defined hypothesis: that quasi-character-level models will outperform 32k-vocabulary-size BPE models at small training set sizes, and tests that hypothesis thoroughly.
- The experiments on domain adaptation, catastrophic forgetting and vocabulary flexibility are potentially useful for practitioners.

Weaknesses:
- The work ignores a substantial body of work on character-level translation; there seems to be an abrupt cut-off at 2017. Here is a sampling for missing references that stood out to me: https://arxiv.org/abs/1610.03017;  https://arxiv.org/abs/1808.09943; https://arxiv.org/abs/1810.01480. Likewise, I think it would be useful to cite some of the recent work on character-level large language modeling with Transformers: https://arxiv.org/abs/2103.06874; https://arxiv.org/abs/2105.13626;  https://arxiv.org/abs/2106.12672. Along these lines, I think that it was already known that smaller vocabularies help low-resource languages. For example, Cherry et al (already mentioned; https://arxiv.org/abs/1808.09943), emphasize that character-level models have their greatest advantage when data sizes are small, and Sennrich and Zhang (https://aclanthology.org/P19-1021/) show that reducing vocabulary size improves truly low-resource NMT.
- The main contribution of this paper seems to be the suggestion that being quasi-character-level allows one to reap the benefits of character-level modeling without paying the computational costs, but this is never demonstrated explicitly. The authors do not try to determine to what degree being quasi-character-level (a) hurts or helps performance with respect to fully character-level; or (b) improves efficiency with respect to fully character-level. A vocabulary-size ablation, graphing speed-versus-quality with a smooth transition of sizes from character to 100 to 400 and then maybe doubling in size thereafter up to 32k would strengthen the work.


**Summary Of The Paper:**

This paper describes a series of experiments to test the effectiveness of quasi-character-level NMT, which they define as subword vocabularies with sizes between 100 and 400 types. These are tested in the context of artificial low-resource scenarios (i.e.: subsampling medium-sized data sets), and in the context of catastrophic forgetting. In this latter experiment, it is shown that character-level models do suffer from forgetting as they are adapted to other tasks, but their more general vocabulary allows them to be adapted more easily.

**Summary Of The Review:**

This paper adds to the ongoing discussion of character-level versus subword-level MT, providing more data points in more architectures indicating the character-level advantage with small training set sizes. It also adds an interesting point about domain flexibility. Overall, the experimental contributions do not seem to be sufficient to warrant a main-conference paper; the results on low-resource scenarios were expected, and the results examining domain flexibility and forgetting are not enough to carry a paper. More experimental emphasis on the utility of quasi-character-level versus character-level, or more experiments on preventing forgetting as suggested in the future work could strengthen the paper substantially.

---

### Official Review · Reviewer_FBrF · 2021-11-02

**Correctness:** 3
**Technical Novelty And Significance:** 2
**Empirical Novelty And Significance:** 2
**Recommendation:** 3
**Confidence:** 5

**Main Review:**

The paper is motivated by an important problem, that of developing a principled method to choose vocabulary sizes for NMT models.

The paper is well-written, easy to follow, but also short on technical details for experiment descriptions.

There are several issues in the experiments that are of immediate concern:

1. The experiments on the low-data regime is done synthetically, i.e. no actual low-resource language pairs were used. Why didn't the authors choose an actual low-resource language pair such as Azerbaijani or Tigrinya? The selected languages all use Latin script only, so the language diversity is also quite low in that regard.

2. The current State of the art NMT systems typically use byte-level-fallback in case of <unk> tokens. This mitigates the effect of all long-tailed tokens degenerating to <unk> even for subword models. Why didn't the authors include this fall-back mechanism in their experiments? Since, the handling of long-tailed tokens is described as one of the primary modeling objectives of this paper, the exclusion of this standard technique seems to be an important omission.

3. The most important concern is that vocabularies of 32K were built using 50K, 100K sentences. This deviates from other works (e.g. https://arxiv.org/pdf/2010.04924.pdf, which studies long-tail in NMT reports 10K vocabulary for training data sizes ranging from 10K to 200K and https://arxiv.org/pdf/2004.02334.pdf, which studies subword vocabulary size shows that larger vocabulary size significantly impacts translation under low-resource settings). What this suggests is that 32K might represent a sub-optimal vocabulary size for 50K and 100K training sentences, and given that this hyperparameter is hugely important for establishing the results, the downstream conclusions can't be trusted unambiguously under this setting. Effectively, by training a 32K vocabulary size on 50K sentence pairs, the vocabulary will degenerate to word-level, rendering subword to quasi-character level comparison flawed and inconclusive. Looking at Figure 1, this looks to be the case actually.

4. Certain claims made by the authors aren't quantitatively qualified vey well. For example, "Quasi-character level transformers achieve better consistency across domains..." isn't quantified: the absolute BLEU values can't be used to make such claims (they are meaningful for system comparisons only), and consistency isn't explicitly measured.

5. Since this is a MT-focused paper, the authors should also try to include comparisons using state of the art neural metrics such as COMET and BERTScore, rather than using surface-level overlap metrics such as BLEU or ChrF.

**Summary Of The Paper:**

This paper inspects the behavior of quasi-character level NMT models, where "quasi-character level" is meant to depict subword vocabularies which are an order of magnitude smaller than typical subword vocabulary sizes (e.g. 32K, 64K) used to train state of the art NMT models.

The dimensions of inspection are:

1. General NMT performance measured by BLEU
2. Impact of Vocabulary size on fine-tuning performance
3. Performance comparisons as a function of training data size
4. Performance measured across different domains and NMT architectures

The experiments show that smaller subword vocabularies could lead to better performance when training data size is small. Another result is that catastrophic forgetting is more prominent when finetuning models with smaller vocabulary sizes. However, there are several issues in the experimental protocol used, which make the presented conclusions in the paper non-definitive.

**Summary Of The Review:**

The paper studies an important problem, but the experimental protocols adopted do not allow any conclusive results. The main claim on the benefit of quasi-character level vocabulary in low-data regime (over a well-selected subword vocabulary size) isn't justified by the presented experiments, due to the issues pointed out above.

---

### Official Review · Reviewer_7Xbu · 2021-11-03

**Correctness:** 1
**Technical Novelty And Significance:** 2
**Empirical Novelty And Significance:** 2
**Recommendation:** 3
**Confidence:** 5

**Main Review:**

Overall, I certainly acknowledge the authors’ effort on open vocabulary problems, which is worthy to explore. However, the author merely shows some old news for the MT community, such as character-level NMT works for extremely low resource settings. In addition, many approaches are designed for extremely large vocabulary problems, such as byte-level encodings, e.g. byT5[1], GPT3[2]. Also, with the increase of data volume, extremely low resources are no longer the mainstream scenario for academia and industries. Strategies that work for the rich resources are more likely to be adopted by the community.

The authors disclose some foundation information, e.g. SELENE project No. 871467 and DeepPattern project No. PROMETRO/2019/121, in acknowledgment, which may violate the double-blind policy.

Reference:

[1] Xue, L., Barua, A., Constant, N., Al-Rfou, R., Narang, S., Kale, M., Roberts, A., & Raffel, C. (2021). ByT5: Towards a token-free future with pre-trained byte-to-byte models. ArXiv, abs/2105.13626.

[2] Brown, T.B., Mann, B., Ryder, N., Subbiah, M., Kaplan, J., Dhariwal, P., Neelakantan, A., Shyam, P., Sastry, G., Askell, A., Agarwal, S., Herbert-Voss, A., Krueger, G., Henighan, T.J., Child, R., Ramesh, A., Ziegler, D.M., Wu, J., Winter, C., Hesse, C., Chen, M., Sigler, E., Litwin, M., Gray, S., Chess, B., Clark, J., Berner, C., McCandlish, S., Radford, A., Sutskever, I., & Amodei, D. (2020). Language Models are Few-Shot Learners. ArXiv, abs/2005.14165.

**Summary Of The Paper:**

The authors revisited open vocabulary problems in translation with a character-based approach. They claimed that segmentation approaches such as BPE may be affected by the vocabulary size, which is heuristic. However, purely character-level segmentation makes the sequence too long, thus making the MT model hard to capture long dependencies.

To achieve a good trade-off, this work explores quasi character level Transformer models for machine translation tasks. They find that the quasi character level MT outperforms their large vocabulary BPE counterparts under the low-resource settings.

**Summary Of The Review:**

Reject

---

### Decision · Program_Chairs · 2022-01-20

**Decision:**

Reject

**Comment:**

This paper demonstrates the hypothesis that a very small word piece vocabulary (giving a "quasi character level" model) outperforms current methods of neural MT in truly low resource scenarios, and provides some auxiliary studies around word piece frequency and domain transfer. It considers LSTM, CNN, and Transformer NMT models. This is useful information for people working in low resource scenarios to know.

The paper got 3 reviews by people with very strong machine translation expertise. There was a general consensus that the paper was insufficiently aware of prior work on this topic and the paper had problems in experiment construction which raised issues about the comprehensiveness of the result. That is, while this paper adopts a more extremely small vocabulary, Sennrich and Zhang (2017) already showed that a much smaller subword vocabulary can give much stronger results for low resource MT (while Araabi and Monz questioned whether this was as true for Transformer NMT. Meanwhile Cherry et al. (2018) and Kreutzer and Sokolov (2018) argued already the benefits of (almost) character-level NMT. On the experimental side, both not having results on genuinely low-resource scenarios and the commented of Reviewer FBrF that the problem with larger subword vocals here may be mainly due to the small corpus size used for constructing the subword vocabulary are both quite important. Moreover, as mainly an MT experimental study, this paper seems better suited to a more specialized audience of MT researchers at an ACL, WMT, AMTA, etc. venue.

I recommend rejecting this paper as not sufficiently novel, with experiments that need further work, and lacking strong interest to a broader representation learning audience.